# Then and Now: A Comparative Historical Toponomastics Analysis of Station Names in 2 of Singapore's Mass Rapid Transit (MRT) Lines

**Shaun Tyan Gin Lim** and **Francesco Perono Cacciafoco** *

Linguistics and Multilingual Studies, School of Humanities, Nanyang Technological University,
48 Nanyang Ave, Singapore 639818, Singapore; l170002@e.ntu.edu.sg
* Correspondence: fcacciafoco@ntu.edu.sg; Tel.: +65-651-483-95

**Abstract:** Public transport is integral to the development of cities. It promotes economic development, mitigates environmental degradation, and fosters a sense of social cohesion. Notwithstanding, one can understand a place's culture, geography, history, languages, and sociopolitical structures by studying the naming practices in public transport, such as bus routes and train stations, among others. This article studies the naming conventions in Singapore's Mass Rapid Transit (MRT) system, which serves millions of commuters daily, and alludes to the importance of public transport in urban spaces. The paper analyses MRT station names, which can be regarded as toponyms, of the North South and Downtown lines according to two aspects: firstly, by conducting a linguistic analysis of the languages used in naming these MRT stations and, secondly, by applying toponymic classifications from current research in grouping the MRT stations themselves. Ultimately, the study compares the naming practices of Singapore's oldest and second newest MRT lines using a sociolinguistic and historical toponomastics mixed methods approach, studying the MRT station names based on social categories as well as using historical sources to account for the linguistic and historical meaning of these toponyms. This work is aimed at providing scholars and a general audience with a better understanding of Singapore's language, culture, and society through the analysis of the naming practices of the MRT station names, unique toponyms in the urban transport of the Lion City.

**Keywords:** toponymy; historical toponomastics; sociolinguistics; urban studies; Singapore; public transport

---

## 1. Introduction

Public transport is vital in urban spaces. A 2009 position paper by the International Association of Public Transport states that a well-developed public transport network

> "enables cities to thrive and fulfil their economic, environmental and social aspirations. Good public transport is vital to successful urban areas, enabling people to access jobs and services, employers to access labor markets and businesses to reach the customers for their services. Good public transport is clean, fuel and carbon efficient and enhances the attractiveness of city centers and the health of the citizens". [1] (p. 1)

Public transport facilitates cities' economic growth by decreasing congestion, raising productivity, connecting labor and wealth to the marketplace, and allowing individuals and businesses to increase their incomes. By reducing the reliance on cars, public transport also decreases carbon emissions and negative environmental impacts of congestion. Such trends are observed in urban areas and developed countries worldwide from Sweden [2] to Germany [3] to Australia [4] to Japan [5] and even in Istanbul,

which is located in developing Turkey [6]. At the social level, taking public transport can lead to greater participation in social activities, thereby building social cohesion [7]. Additionally, public transport allows its users to empathize with the city's complexities and experience its sociocultural and linguistic diversity [8].

Naming policies utilized in public transport and built spaces reflect a city's culture, history, geopolitical realities, and sociolinguistic landscape, especially in multilingual societies [9–12]. This is hardly surprising, since toponyms, or place names, can be used to understand how inhabitants of a place compartmentalize and interact with their geospatial, sociopolitical, and cultural environments [13].

Some research has also been conducted on how toponyms, especially the names of public transport systems, are used as branding exercises [14–17]. A few local authorities have sold their naming rights of public transport to major corporations. This would boost the revenue of cities, enabling them to invest more in transport while offering these companies marketing opportunities to brand their organizations and products on board these transport systems. Light and Young [16] cite, among others, the examples of Tampa, Florida, where the TECO Line Streetcar System was named after a utilities company and the Atlantic Avenue–Barclays Center station in New York, which was renamed after the Barclays Bank as part of a US$4 million sponsorship deal. In another example, Dubai's Roads and Transport Authority (RTA) sold the naming rights of the former Sharaf DG Metro Station to Mashreq Bank in 2020 [18]. Accordingly, the station is now known as the Mashreq Metro Station. Furthermore, two Dubai Metro stations were renamed: Palm Deira and Nakheel Harbour and Tower to Gold Souq and Jabal Ali, respectively.

Recent toponymic research illustrates how, in some cases, the commodification of toponyms has even extended to sporting clubs situated in the locality. Creţan [19] uses the example of Poli Timişoara Football Club in Timişoara, Romania. Initially, the local government sold the naming rights of the team to increase their coffers. However, when the club's private owners could not support the club financially, the local authorities reacquired the naming rights of Poli, now rebranded as a new and smaller football team, Asociaţia Club Sportiv Poli. Creţan's research shows how commodifying names can evoke strong reactions among parties who "own" the name, leading to social inequalities and polarization.

Other scholars have also focused on the politics of naming practices in urban spaces. As part of the modern political culture, street names, which are part of urban transport systems, are named (and sometimes renamed) to commemorate important historical figures and/or events [20–22]. Toponyms thus legitimize the sociopolitical order of the day. As Azaryahu [21] notes, one can witness the intersection of hegemonic political and ideological discourse with the spatial practices of everyday lives via commemorative street names. These toponyms are therefore tools to justify and perpetuate the state narrative of historical events in the hearts and minds of their populace. These findings add value to the larger category of critical toponymy, which studies toponyms in the wider sociopolitical context because place naming is affected by political and cultural decisions [23–25].

Notwithstanding, public transport is also extremely important in Singapore, a sprawling metropolis. Its public transport system is ranked among the best in the world [26] and serves over 7.54 million passengers daily [27]. However, the links between public transport and toponyms in Singapore have not been well documented as most research on Singaporean public transport centers on detailing and evaluating policies of the past, present, and future [28–31]. Meanwhile, scholarly work on toponymy in twenty-first-century Singapore largely focuses on street names [32–36]. The most recent work, Cavallaro, Perono Cacciafoco, and Tan's paper, applied the Sequence Occupance Theory [36] to ten micro-toponyms in Singapore. The Sequence Occupance Theory, associated with Derwent Whittlesey, argues that successive civilizations leave their cultural imprints on a place, each contributing to the cumulative cultural landscape (and toponymy). The authors conclude that these place names bear the indelible influence of all cultures that have "sequentially" occupied Singapore from past to present.

This article analyzes Singapore's Mass Rapid Transit (MRT) station names of two lines: the North South Line and the Downtown Line. Given that the paper seeks to study Singapore's MRT station names over time, the North South Line, Singapore's oldest MRT line, and the Downtown Line, Singapore's second newest MRT line, are instructive for this purpose. The Downtown Line is selected over the Thomson–East Coast Line since only three stations are operating in the latter as of the time of publication.

The paper introduces Singapore and her languages as well as current methods of toponymic research in Singapore. Thereafter, a brief history of the MRT system and naming practices of MRT stations will be provided. Finally, the authors will conduct a linguistic analysis of the 61 MRT station names of the North South and Downtown lines. Specifically, the authors will analyze the languages used in naming the MRT station and adapt and apply existing toponymic classifications of these station names. Apart from identifying trends in the languages and dialects (e.g., English, Mandarin Chinese, Malay, Tamil, Hokkien, Teochew, etc.) used in naming the MRT stations, the authors aim to contrast linguistic naming practices in older (North South Line) and newer (Downtown Line) public rail networks in Singapore, thereby adding a comparative approach to toponymic studies in Singapore.

## 2. An Introduction to Singapore and Singapore's Languages

Singapore is an island nation located at the southern tip of the Malay Peninsula in Southeast Asia, sandwiched between Malaysia to the north and Indonesia to the south. The country is an oft-cited example of a well-planned and model city-state [37–39]. Singapore is also described as a "melting pot of cultures"; of Singapore's 4 million residents, 74.3% are Chinese, 13.4% are Malay, 9.0% are Indian, and 3.2% are classified as other [40].

Singapore is also a multilingual nation, with a plethora of languages spoken locally. Table 1 shows the languages most frequently spoken at home by Singaporeans across time.

**Table 1.** Speakers of the main languages in Singapore based on "the language most frequently spoken at home" (%) [41,42].

| Language/Year | 1957 | 1980 | 1990 | 2000 | 2010 | 2015 |
|---|---|---|---|---|---|---|
| English | 1.8 | 11.6 | 18.8 | 23.0 | 32.3 | 36.9 |
| Mandarin | 0.1 | 10.2 | 23.7 | 35.0 | 35.6 | 34.9 |
| Chinese dialects | 74.4 | 59.5 | 39.6 | 23.8 | 14.3 | 12.2 |
| Malay | 13.5 | 13.9 | 14.3 | 14.1 | 12.2 | 10.7 |
| Tamil | 5.2 | 3.1 | 2.9 | 3.2 | 3.3 | 3.3 |
| Others | 5.0 | 1.7 | 0.7 | 0.9 | 2.3 | 2.0 |

While English is the lingua franca in present-day Singapore, few people spoke the language 50 years ago. Only 1.8% of Singaporeans spoke English in 1957. English was the language of the British, who colonized Singapore from 1819 to 1959, and its main role was to produce English-speaking colonial administrators [43]. Despite having a majority Chinese population, only 0.1% of the population spoke Mandarin in 1957. It is worth noting that the forefathers of Chinese Singaporeans today sailed down from southern China during the nineteenth and twentieth centuries. These immigrants, although being ethnically Chinese, spoke Chinese dialects commonly associated with southern China, such as Hokkien, Teochew, Cantonese, Hainanese, and Hakka. Hokkien was spoken amongst speakers of different Chinese dialects, while Bazaar Malay (a variety of the Malay language) was commonly used amongst Singaporeans of different ethnicities [44].

The sociolinguistic situation in Singapore began to change after the country gained independence in 1965. In 1966, the government introduced the bilingual education policy. Singaporean students were required to learn both the English language and their official ethnic language as a "Mother Tongue Language" [45]. The rationale for the bilingual education policy was illustrated by the then Education Minister, Tony Tan. In a statement in 1986, the minister argued that

"Children must learn English so that they will have a window to the knowledge, technology, and expertise of the world. They must know their mother tongues to enable them to know what makes us what we are". [46] (p. 84)

Furthermore, in 1979, the government launched the Speak Mandarin Campaign, aimed at getting Chinese Singaporeans to eschew dialects for Mandarin Chinese [47]. Consequently, more Singaporeans spoke English, rising from 1.8% (1957) to 36.9% (2015). Since Chinese Singaporeans, regardless of dialect group, had to study Mandarin Chinese in schools in post-independence Singapore, the percentage of Chinese users increased greatly. The increase in Chinese speakers has simultaneously coincided with a fall in the number of Singaporeans using Chinese dialects, which are today spoken largely by older Singaporeans.

The use of Malay and Tamil have seen small declines; Malay was the most frequently spoken language at home for 13.5% of Singaporeans in 1957. In 2015, this figure dropped to 10.7%. The percentage of Singaporeans who spoke Tamil dropped from 5.2% in 1957 to 3.3% in 2010. This could also be due to the success of the bilingual education policy, which has led to Malay and Indian Singaporeans speaking English at home rather than Malay and Tamil, respectively.

## 3. Existing Toponymic Classifications in Singapore

Toponymic research in Singapore over the past 20 years has sought to categorize Singapore's place names, especially street names, into viable groups and, subsequently, analyze naming trends of Singaporean toponymy. The following section documents three of these notable studies and assesses the strengths of these works, especially in implementing these toponymic classifications in this study.

### 3.1. Savage and Yeoh (2003; 2013)

Geographers Victor Savage and Brenda Yeoh, in 2003 (updated in 2013), published *Toponymics: A Study of Singapore Names* [32,33]. The authors note that Singapore's toponyms were organized into seven categories:

1.  Colonial names derived from important places and people in the British Empire.
2.  Malayanized names after places, flora, and fauna. These names are usually from the Malay language.
3.  Names associated with different racial groups.
4.  Names after prominent Asian leaders, wealthy landowners, and other people.
5.  Descriptive names expressing topographical features and landmarks or indicating former or current land uses, trades, and activities.
6.  Numerical names, as found in many Housing Development Board (HDB) towns.
7.  "Themed" names associated with opera, colors, or local bird species.

This system is neat and easy to use when organizing Singapore's street names and MRT station names. However, the authors decided against adopting this classification system in its entirety for a few reasons. Firstly, the linguistic analysis of MRT station names, which the authors aim to achieve in this study, must include all languages and dialects spoken in Singapore, not just Malay. Besides, street names named after "important people in the British Empire" and names linked to "prominent Asian leaders, wealthy landowners, and other people" can be grouped under the broader category of "eponymous" names that come from the name of a person. Furthermore, "colonial names derived from important places in the British Empire", "Malayanized names after places, flora, and fauna", and "names associated with different racial groups" can be regarded as "borrowed" names that are derived from a foreign language, culture, or concept. Names that denote topographical features and landmarks can be regarded as associative names instead of descriptive ones [48,49]. Finally, numeral names and "themed" names are rare amongst MRT station names.

### 3.2. Ng (2017)

In *What's In The Name? How The Streets And Villages In Singapore Got Their Names* [34], Ng Yew Peng argues that the book is the first of its kind that analyses naming trends of Singapore's streets and villages. Under Section 3, Ng lists 11 ways of labeling Singapore's place names:

1. Roads named after foreign places, including Britain and the British Empire, Malaysia, Indonesia, India, China, and Burma.
2. Anthroponyms named after nonresidents (British, Europeans, and Asians who did not live in Singapore) and residents (Europeans, Chinese, Malays, South Asians, Arabs, and Jews who lived in Singapore).
3. Places named after the main economic activity conducted there, such as entrepot trade and agriculture.
4. Places named due to activities of war.
5. Places named after the political system.
6. Places named after historical sites.
7. Places associated with religion.
8. Place names linked to families.
9. Places linked to education.
10. Place names named after females.
11. Place names with a "theme".

Ng's taxonomy is comprehensive and accounts for the origins and development of Singapore's street names. However, in the case of Singapore's MRT system, which numbers over 130 stations today [50], this arrangement is cumbersome and confusing and, hence, cannot be applied fully. Certain groups, such as "anthroponyms", "place names linked to families", and "place names named after females", can be merged under "eponymous" place names as mentioned in Section 3.1.

### 3.3. Perono Cacciafoco and Tuang (2018)

Perono Cacciafoco and Tuang's 2018 article "Voices from the Streets: Trends in Naming Practices of Singapore Odonymy" [35] analyzed the naming practices of 150 street names from three of Singapore's official languages (English, Chinese, and Malay). The authors employed the following six classes to categorize the odonyms:

1. Commemorative naming after prominent figures or significant events.
2. Borrowed toponyms from foreign places or languages or from racial groups.
3. Thematic toponyms connected with a specific area (in the case of Bukit Timah in central Singapore, some streets are named after trees).
4. Descriptive toponyms that refer either to landscape features, such as the morphology of roads, or toponyms derived from facilities or sceneries existing in their areas.
5. Place names derived from trees', plants', or flowers' names.
6. "Others" that comprise places named after specific designations, objects, animals, or fruits with no clear specifications of descriptive origins.

These six classifications are suitable for grouping place names on a smaller scale, such as the MRT station names in Singapore. However, Perono Cacciafoco, and Tuang could do more to clarify why a separate category was created for toponyms derived from the names of trees, plants, and flowers when this category "could also serve descriptive functions" [35] (p. 18). Additionally, "toponyms derived from facilities or sceneries existing in their areas" have been argued to be an associative naming strategy rather than a descriptive one [48,49]. It is also worth noting that thematic place names, as aforementioned, are uncommon amongst MRT stations.

*3.4. The Proposed Toponymic Classification*

After analyzing the existing literature used to classify place names in Singapore, especially Singapore's streets, the authors worked with, adjusted, and improved on these criteria, recommending seven groupings for evaluating the naming trends of Singapore's MRT station names. These categories would be utilized by the authors to investigate the naming practices and trends of MRT station names in the North South and Downtown lines.

1.  Associative MRT names related to nearby topographical features, landscapes, landmarks, facilities, and sceneries, be it physical (for instance, a hill or water body) or man-made (like roads, tourist attractions, industrial estates, etc.).
2.  Borrowed names from foreign languages, cultures and places or from racial groups.
3.  Descriptive MRT station names that elucidate qualities of the particular station or area, be it the activities, economic trades, and land uses or the trees, plants, flowers, and crops grown in the locality. In some cases, descriptive MRT names also describe traits of the station's area.
4.  Eponymous naming, where the station name is named after important British, European, and Asian leaders and businessmen (and their families) or women who contributed to Singapore's development.
5.  MRT stations named after legends and anecdotes.
6.  Occurrent names christened after famous historic events.
7.  "Other" MRT station names whose development and origins are largely unknown.

## 4. An Introduction to the MRT System and its Naming Practices

*4.1. The Great MRT Debate*

The MRT was first mooted as part of the State and City Planning Project, formed in 1967 by Singapore's State and City Planning Department. The project, which included local planners and experts from the United Nations Development Program, sought to chart Singapore's future housing, transport, industrial, and infrastructure development. The State and City Planning Project led to the Singapore Ring Concept Plan in 1971, which recommended a rail-based MRT system to serve the densely populated city area and facilitate island-wide transportation [51–54].

However, the MRT system was not approved and built overnight. The high cost of the MRT system forced the government to adopt a cautious approach; in 1972, it was estimated that an MRT system would cost Singapore SGD $400 million in capital expenditure alone [55]. This entailed a trade-off: spending on an MRT system meant that there were fewer financial resources available for other public goods like housing, defense, and education for the newly independent country; thus, the government was unwilling to commit immediately to constructing the MRT system.

In the 1970s, the government commissioned a series of studies to evaluate the feasibility of the proposed public rail network. These included studies by the American consultancy Wilbur Smith and Associates (1972), the World Bank (mid-1970s), and a team from Harvard University (late 1970s). Besides reports done as part of the three-part Mass Transit Study, ministries like the Ministry of National Development (1976) and the Ministry of Defense (1978) deliberated on various aspects of having an MRT system. However, opinions were divided and no conclusive decision was reached on whether Singapore should build an MRT [56].

A strong proponent of the MRT system was Ong Teng Cheong, who, in 1978, became the Minister for Communications. Ong's portfolio oversaw Singapore's public transportation then. Ong was trained as an architect and was also part of the 1967 project that first suggested the MRT system. At a National Day dinner in his constituency, Ong made his case for the MRT and drew comparisons with other metropolises with established rail networks:

"Certainly, the MRT can contribute to the growth and vitality of a city and allow for more intensive development and better economies of scale. New York, London or Tokyo would

not be able to support the amount of urban and economic activity that they do today without their subway systems . . . . Apart from its mass transportation capability, MRT would also represent a quantum leap in the quality of public transport. The comfort, efficiency and reliability of MRT would make urban travel less burdensome, and enable us to make more meaningful use of our time". [57] (p. 1)

However, Ong was strongly opposed by others, including members of his own government. Then Deputy Prime Minister Goh Keng Swee was worried about the cost of building an MRT system and preferred the all-bus system recommended by the Harvard University team. Trade and Industry Minister Tony Tan was famously quoted as saying, at a forum at the National University of Singapore, that it was "foolish to build MRT" [58]. In October 1980, a 75-min debate was broadcast on television, featuring both sides in the MRT debate: the Harvard team, which pushed for an all-bus system in Singapore, and Wilbur Smith and Associates, representing the pro-MRT camp. This debate came to be known as "The Great MRT Debate". The question of whether an MRT system should be constructed in Singapore also generated great interest from ordinary Singaporeans; eight forum letters were written about the MRT debate between September and October 1980.

The turning point came in 1982 after the government's Comprehensive Traffic Study found that an all-bus system was unpractical by 2000 and recommended the immediate construction of the MRT system [59]. In May 1982, the government decided to build and finance the SGD $5 billion MRT system. As Minister Ong would announce:

"The Government has now taken a firm decision to build the MRT. The MRT is much more than a transport investment, and must be viewed in its wider economic perspective. The boost it'll provide to long term investors' confidence, the multiplier effect and how MRT will lead to the enhancement of the intrinsic value of Singapore's real estate are spin-offs that cannot be ignored." [56]

The announced MRT system was 67 km long and comprised three lines. Construction of the North South Line began first as this was regarded as the route most badly hit by traffic congestion. After five years, the MRT system began operations. On 7 November 1987, five stations began operations: Yio Chu Kang, Ang Mo Kio, Bishan, Braddell, and Toa Payoh. *The Straits Times* reported that 57,000 people visited Toa Payoh and Braddell stations over the opening weekend [60], a testament to Singaporeans desiring to experience the novelty of rail transport in their city. On 12 December 1987, nine more stations were added. This allowed Singaporeans to travel from Novena to Raffles Place on the North South Line and from Outram Park to Raffles Place on the East West Line, Singapore's second oldest MRT line. By 1988, more MRT stations from Tiong Bahru to Clementi were opened.

More MRT stations were added in 1996 to the North South Line, serving Singaporeans who lived in the north and northwestern part of the island. In 2003, Singapore's first automated MRT line, the North East Line, opened. Subsequent MRT lines, such as the Circle Line and Downtown Line, are also automated.

Singapore's sixth and newest MRT line, the Thomson–East Coast Line, will be opening in stages from 2019. Currently, only three MRT stations, Woodlands North, Woodlands, and Woodlands South, are in operation. With more MRT lines, such as the Jurong Region Line and Cross Island Line, under construction, Singapore's public transportation by 2030 will be more extensive and accessible, comprising some 280 MRT stations, double the number of today's stations, and placing eight in 10 homes within a 10-minute walk to an MRT station [61].

*4.2. A More Decentralized Approach to Naming?*

In the early days of the MRT system, the Provisional MRT Authority, formed by Minister Ong Teng Cheong in 1980, was responsible for deciding the sites and names of the proposed MRT system. After the government approved the plan to build the MRT, the MRT Corporation replaced the Provisional MRT

Authority in 1983 and oversaw the above-mentioned roles, along with building and operating the MRT system. In 1995, the Land Transport Authority (LTA, located at Hampshire Road near Little India MRT station, Singapore) was established. The LTA plans, designs, builds, and maintains Singapore's land transport networks. The LTA also oversees and proposes names for new MRT stations, although they have occasionally modified these names. This is seen in the case of the former Singapore Polytechnic MRT station, later renamed by the LTA as Dover MRT station, because the public felt that they should not subsidize the cost of building this MRT station, which primarily served students.

For newer MRT lines, such as the Downtown Line, the LTA has adopted a more consultative approach. The LTA firsts suggests working names for MRT stations on newer MRT lines. For example, in 2008, the LTA proposed preliminary names for nine stations on the Downtown Line: Rochor, Stevens, Duchess, Sixth Avenue, Blackmore, Beauty World, Hillview, Cashew, and Petir. The LTA then asked members of the public to suggest names for these stations. Singaporeans could either maintain the working name or recommend new names for the stations and provide justifications for their choice. According to the LTA, station names should "help commuters easily remember and identify the area of the station" and "reflect the history and heritage of the station surroundings" [62]. The LTA will then shortlist names for these MRT stations. In the above instance, the LTA chose 27 names from more than 3000 names [63]. The public will then vote for their preferred station names. Finally, the LTA will compile the results of the public vote before sending the MRT station names to the Street and Building Names Board for their final approval. The shift away from a top-down naming strategy to one that calls upon the public to suggest possible station names and vote on them reflects a more consultative and cooperative approach whereby the LTA and government partner with Singaporeans in shaping the names and identity of MRT stations.

## 5. Methodology

Since the paper aims to study Singapore's MRT station names over time, the authors will analyze MRT station names from the North South Line, Singapore's oldest MRT line, and the Downtown Line, Singapore's second newest MRT line as the newest MRT line in Singapore only has three stations running. A total of 61 MRT station names from the North South and Downtown lines were collected as a sample for analysis. Of these, one station (Newton) is an interchange serving both lines and, thus, would appear twice in the analysis of MRT station names. These station names were first recorded from the official Land Transport Authority links for North South Line [64] and Downtown Line [65]. The authors then comprehensively analyzed these toponyms according to the language and naming strategy behind the MRT station names.

To ensure the accuracy of the analysis, the authors counterchecked against a range of primary and secondary sources, including maps, digitalized newspapers, and established scholarly sources on Singapore toponymy [33,34] as well as official government sources, including the National Heritage Board, the Urban Redevelopment Authority, and the National Library Board, which also contain information behind the origins and development of place names in Singapore.

The data were then analyzed and presented using pie charts to compare the language used in naming the MRT station names for both MRT lines. Figures were also generated to group the station names according to their naming practices. The authors have, using existing toponymic literature from Singapore and beyond, aimed to organize the MRT station names into the following categories:

- Associative MRT names related to a nearby natural or anthropic landscapes.
- Borrowed names from foreign languages, cultures, or racial groups.
- Descriptive MRT station names that describe traits (e.g., land uses or formerly cultivated crops) of the station or the place the MRT station serves.
- Eponymous naming, where the station name is named after famous people.
- MRT stations named after legends and anecdotes.
- Occurrent names derived from famous historic events.

- "Others": MRT station names with no clear origins.

The study uses a (socio)linguistic and historical toponomastics approach to investigate the languages used and naming trends of MRT station names in Singapore. Sociolinguistics describes (place) names with reference to social categories, such as "male", "female", "young", "old", "native", and "migrants", among others [66]. The authors, as part of the aforementioned historical toponomastics method, use historical documents to conduct a linguistic and historical interpretation of the toponym, similar to most historical toponomastics approaches [67]. The authors have used the oldest MRT line, the North South Line, and the second newest MRT line, the Downtown Line, in this study. The authors opted for the Downtown Line since the sixth and newest MRT line in Singapore, the Thomson–East Coast Line, has only three stations in operation as of August 2020. Through analyzing a newer and an older MRT line, the authors aim to compare and contrast strategies used in naming MRT stations, be it the languages or naming methods of MRT stations then and now, as well as account for possible differences in naming strategies, thus adding a comparative dimension to a specific field of toponymy in Singapore. The results were then tabulated in Tables A1 and A2.

## 6. Results and Discussion

### 6.1. Languages Used in Naming MRT Stations

As seen in Figure 1, the 27 stations on the North South Line exhibit great linguistic diversity in the languages used in naming these stations. Eleven stations have English names. There are also three stations with Chinese names and four stations with Malay names. Some MRT stations on the North South Line are named using dialects of Mandarin Chinese. Teochew names include Choa Chu Kang ("the river settlement where the Choa clan settled"), Yew Tee ("oil pond"), and Yio Chu Kang ("the river settlement where the Yeo/Yio clan settled"). Ang Mo Kio ("red hair bridge") referred to the bridge built by the British colonial rulers, who were pejoratively known in Hokkien as "red hair devils". There are also stations with names from other languages. For example, Khatib MRT station is borrowed from an Arabic word that refers to a person who delivers the sermon during Friday prayers in the mosque. Novena MRT station is named after a nearby church, Novena Church of Saint Alphonsus. Novena is, in turn, derived from a Latin word, novem ("nine"), or a devotion comprising nine days of prayers or services [68]. Furthermore, Dhoby Ghaut MRT station has Hindi origins, named after the Indian washer men who washed the clothes in a nearby river [33]. Finally, two MRT stations, Jurong East and Toa Payoh, contain two languages in their names.

In Figure 2, it is observed that of the 34 MRT stations on the Downtown Line, English is used to name half of them (17 stations). Malay is the second most common language, with nine stations on the Downtown Line containing Malay names. Only one station, Tan Kah Kee, has a dialectal (Hokkien) name. The station is named after Tan Kah Kee, a Hokkien Chinese businessman and philanthropist who founded Hwa Chong Institution, where the MRT station is located. This is the only MRT station on the Downtown Line to adopt a person's full name as the station's name. Five MRT stations have two languages, Malay and English, in their names. They are Bedok North, Bedok Reservoir, Tampines West, Tampines East, and Upper Changi, respectively. Finally, it is unclear which languages are used to name two MRT stations: Bendemeer and Mattar. Bendemeer MRT is named after the nearby Bendemeer Road, which was named after the residence (Bendemeer House) of Seah Liang Seah, the prominent pepper and gambier trader, who lived in the area. Seah renamed the house to Bendemeer House after he bought the property from the original owner, Whampoa Hoo Ah Kay [33]. While it is unclear what Mattar means or the language it is derived from, many roads near Mattar MRT station are named after plants and in the Malay language. Examples include Jalan Melati ("jasmine" in Malay), Jalan Anggerik ("orchid" in Malay), and Jalan Raya (named after Bunga Raya or "hibiscus" in Malay) [69].

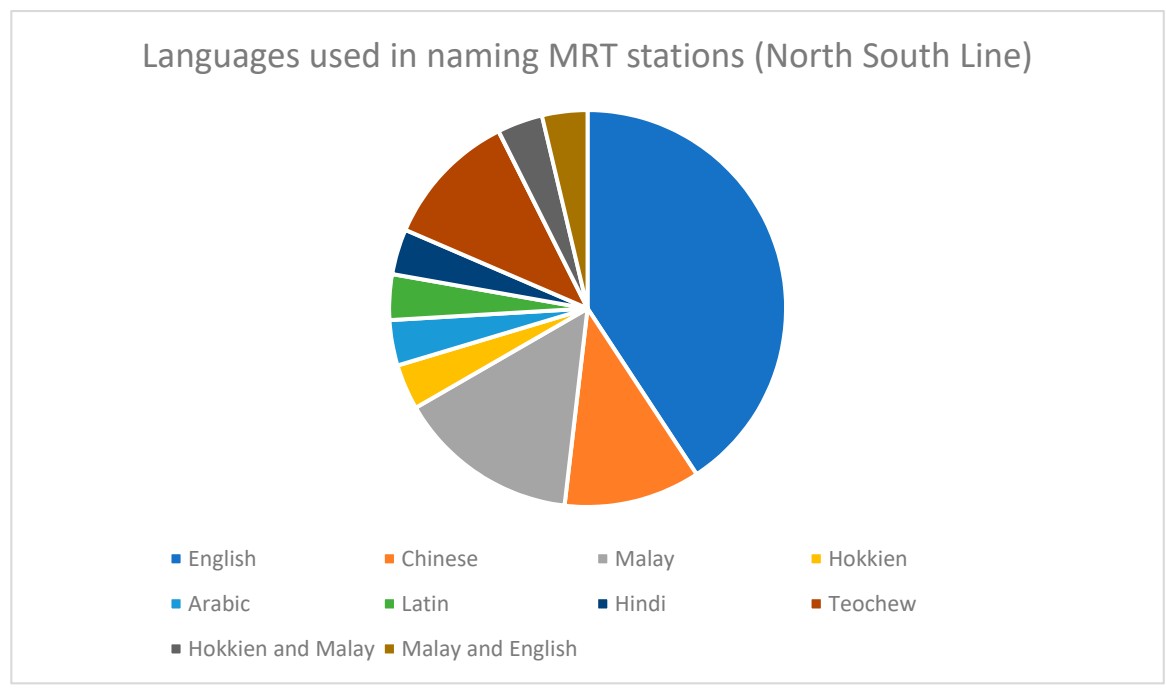

**Figure 1.** Languages used in naming the Mass Rapid Transit (MRT) stations on the North South Line.

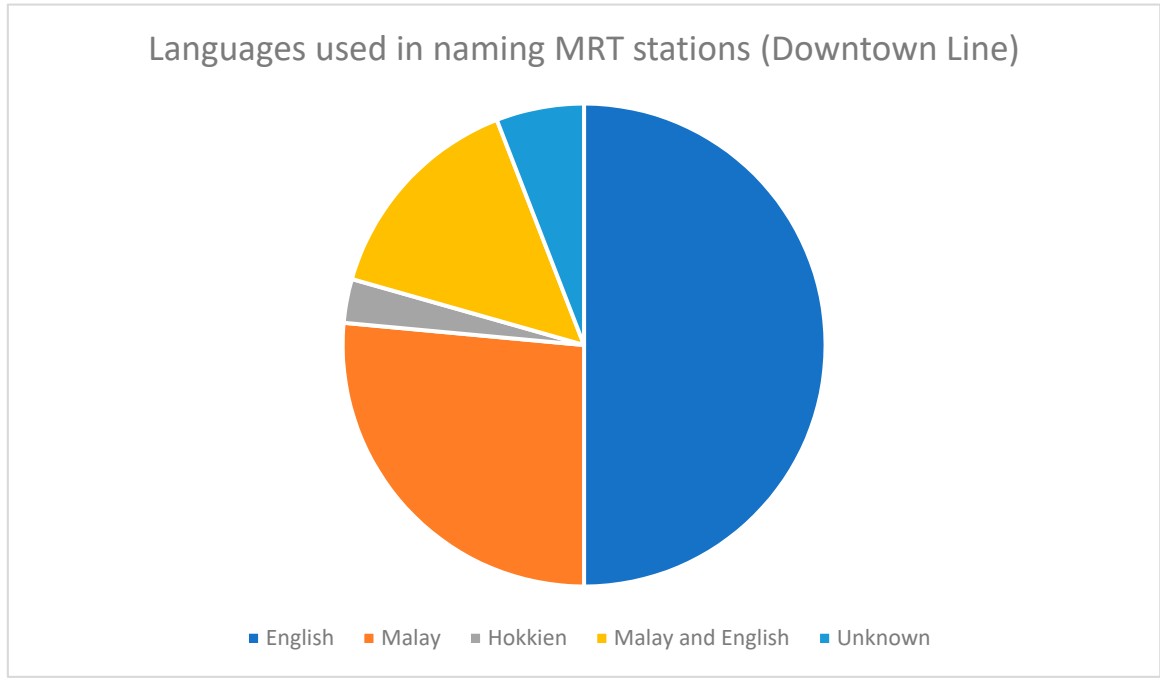

**Figure 2.** Languages used in naming the MRT stations on the Downtown Line.

Taken together, English remains the most common language used to name MRT stations on both the North South and Downtown lines. Malay is the second most prevalent language on both lines. Both the North South and Downtown lines also feature the presence of two languages in a single MRT station (two stations, or 7.41% of stations, on the North South Line and five stations, or 14.71% of stations, on the Downtown Line). However, dialectal MRT stations have fallen sharply: four MRT stations (or 14.8% of stations on the North South Line) are named using dialects. In contrast, the only station with a dialectal name on the Downtown Line is Tan Kah Kee (or 2.94% of stations on the Downtown Line). The fall in dialects used in naming MRT stations reflects the language situation in

Singapore, where fewer locals use dialects. Furthermore, while the North South Line has three MRT stations (or 11.1% of stations on the North South Line) with Chinese names, the Downtown Line has none. Finally, it is noteworthy that MRT stations on the North South Line exhibit greater linguistic diversity as foreign languages like Arabic, Latin, and Hindi are used to name MRT stations, which is not observed in the Downtown Line.

*6.2. Naming Practices behind MRT Station Names*

As evidenced in Figure 3, descriptive naming is the most common naming strategy used for MRT stations on the North South Line. The names of nine stations on the line described their land use and/or trees/crops cultivated in the area and/or area-specific qualities. This was followed by eponymous naming, which featured in seven station names. Four associative MRT stations were named after nearby physical or man-made landmarks while another four MRT stations had borrowed names. The sole station to be named after a historic event, City Hall, celebrates Singapore being granted city status in 1951. The origins of two other names, Jurong East and Bukit Batok, are unknown and thus are placed under "Others".

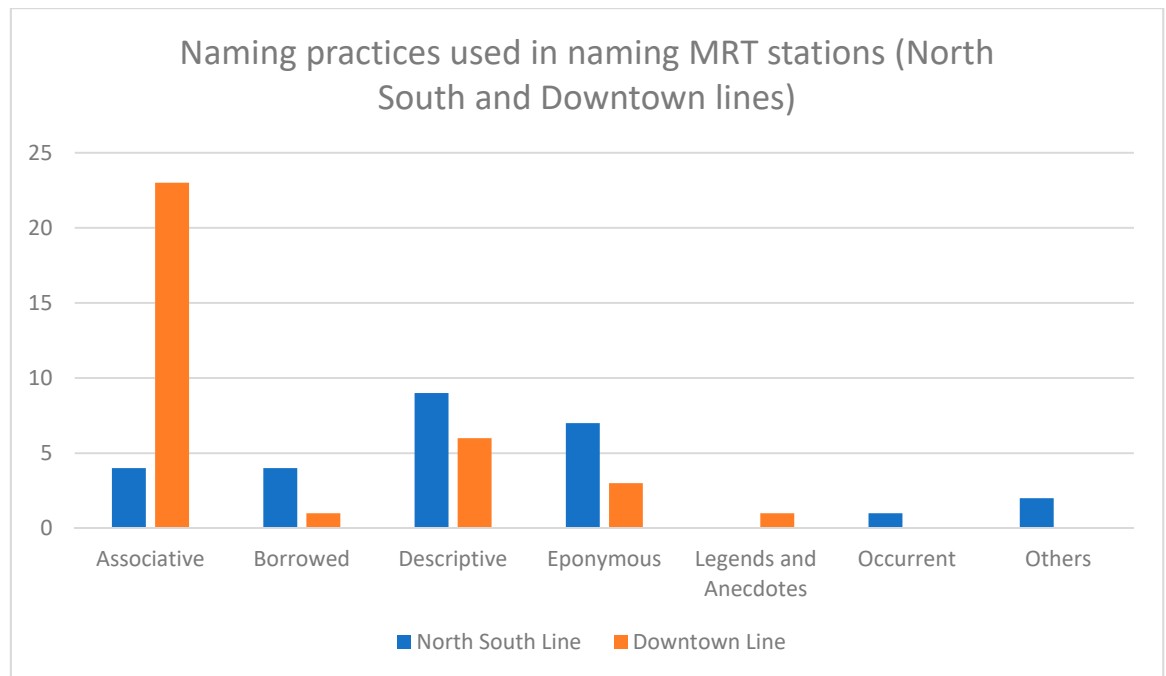

**Figure 3.** Naming practices used in naming the MRT stations on both lines.

In comparison, an overwhelming majority of MRT stations (23 stations) on the Downtown Line utilize associative naming (see Figure 3). Descriptive naming is found in six stations while three MRT stations are named after famous people, reflecting an eponymous naming approach. Bugis is the only MRT station on the Downtown Line to have a borrowed name: it is named after the Bugis people, a local ethnic group. Rochor, the standalone station named after legends and anecdotes, has two possible explanations for its name, both of which are legends. One account of the origins of Rochor is based on Chinese folklore claims that Rochor derived from Wu Zhu, the founder of the Heaven and Earth Society who wanted to overthrow the Qing Dynasty. Wu sailed up the Rochor River and settled on its banks. Rochor was thus named according to the river. The second links the name to Cho Ah Chi, a carpenter who was on the *Indiana*, the expedition led by Sir Stamford Raffles, which resulted in the founding of modern Singapore in 1819. Cho was a member of a triad in Penang and set up a branch in Rochor, giving the district its name [33].

Observations from the above data prove that MRT stations on newer lines show an inclination for associative naming. This is witnessed even amongst newer stations added to older MRT lines. The East West Line, which was constructed around the same period as the North South Line, added four MRT stations to the line in 2017: Gul Circle, Tuas Crescent, Tuas West Road, and Tuas Link, all of them named after nearby roads. Furthermore, two operational stations (Woodlands North and Woodlands South) of Singapore's newest line, the Thomson–East Coast Line, are also named after nearby roads, congruent with the associative naming strategy. Given that the names of newer MRT lines like the Downtown Line are put to a public vote, one possible argument as to why most stations have associative names is because Singaporeans can easily associate and identify the area of the MRT station with roads and other physical and/or anthropic landmarks and, thus, prefer associative names.

## 7. Conclusions

The paper aimed at identifying trends in the languages and dialects, as well as naming strategies, used in the naming of MRT stations. The authors took a comparative approach by looking at MRT stations on the oldest and second newest MRT lines in Singapore. In terms of languages used in naming MRT stations, the results show that English remains the top language in naming MRT stations, perhaps to reflect the language's status as a lingua franca in the island city-state. Malay, being the national language, comes in at second place. However, MRT stations with dialect names have decreased over time, mirroring the language situation in Singapore, which presently has fewer dialect speakers.

The naming practices of newer and older MRT stations have illustrated a shift from descriptive to associative naming. While some names of Downtown Line MRT stations describe features of the station and/or area, over half of the station names on the Downtown Line adopt associative names. As aforementioned, commuters, who now have a say in choosing station names, might prefer associative names that readily link the MRT station with surrounding landscapes.

This article also highlights the interdisciplinary approach in the study of Singaporean toponymy, involving society, culture, sociolinguistics, language policies, and history, all of which leave indelible impacts on the language choice and naming strategy for public transport names. Ultimately, this article lends greater clarity and completeness to toponyms and naming strategies in Singapore and demonstrates the intersectionality between public transport and languages, using toponyms as "linguistic tools". Further research could be done on toponymy in the Singaporean context using a comparative approach. Additionally, it would be interesting to conduct research on the perception Singapore citizens have of their MRT system and the names of the stations. This would provide insights on commuters' opinions towards naming policies of MRT stations (given that Singaporeans are now actively involved in naming newer lines by voting from a list of suggested names, which was previously not done) and their linguistic sensitivity about the choice of MRT names, marrying a variety of disciplines, such as transport, toponymy, sociolinguistics, language policies, psychology, history, and culture together, as this paper has demonstrated.

**Author Contributions:** Conceptualization, S.T.G.L. and F.P.C.; writing—original draft preparation, S.T.G.L.; writing—review and editing, F.P.C.; research, S.T.G.L.; supervision, F.P.C.; project administration, F.P.C. All authors have read and agreed to the published version of the manuscript.

**Funding:** This research received no external funding.

**Conflicts of Interest:** The authors declare no conflict of interest.

## Appendix A North South Line

Table A1. Analysis of MRT Stations on the North South Line.

| Station No. | Station Name | Language | Naming Practice (Strategy Is Underlined) |
|---|---|---|---|
| NS1 EW24 JE5 | Jurong East | Malay English | The roots of the word Jurong are uncertain. It may derive from the Malay words jerung (a voracious shark), jurang (a gap or gorge), or penjuru (corner) (others: unknown origins and development). |
| NS2 | Bukit Batok | Malay | Bukit "hill" + batok "cough". There are many explanations, but they are difficult to verify (others: unclear origins). |
| NS3 | Bukit Gombak | Malay | Bukit "hill" + gombak "a collection of something". Refers to the two hills in Bukit Gombak (associative: hills). |
| NS4 *BP1* JS1 | Choa Chu Kang | Teochew | Named after the Choa clan (eponymous), which settled near the Berih River. The river settlement where the clan settled was known as Chu Kang in Teochew. |
| NS5 | Yew Tee | Teochew | Derived from the Teochew term "oil pond" as the village used to have some oil storage facilities during World War Two (describes the land use in the area). |
| NS7 | Kranji | Malay | Named after kranji trees that were commonly grown in the area (describes trees grown in the area). |
| NS8 | Marsiling | Chinese | Lim Nee Soon, who owned the estate, named it after his hometown of Maxi village in Teochew prefecture (borrowed from a Chinese place name). |
| NS9 TE2 | Woodlands | English | Derived from the rubber trees that flourished in plantations there in the past, resembling woods (describes the quality of the area). |
| NS10 | Admiralty | English | Named after the Admiral of the British Navy, whose residence was located in this area (eponymous). |
| NS11 | Sembawang | Malay | Named after the sembawang tree that grew in the area (describes trees grown in the area). |
| NS12 | Canberra | English | Named after Australia's capital, Canberra (borrowed from Australia's capital). |
| NS13 | Yishun | Chinese | A pinyinized version of Nee Soon, named after the pineapple king, Lim Nee Soon (eponymous). |
| NS14 | Khatib | Arabic | An Arabic word (خطيبkhaṭīb) (borrowed from an Arabic word) meaning a person who delivers the sermon during Friday prayers in the mosque. |
| NS15 | Yio Chu Kang | Teochew | Named after the Yeo/Yio clan who controlled the river (eponymous). The river settlement where the clan settled was known as Chu Kang in Teochew. |

**Table A1.** *Cont.*

| Station No. | Station Name | Language | Naming Practice (Strategy Is Underlined) |
|---|---|---|---|
| NS16 CR11 | Ang Mo Kio | Hokkien | Ang "red" + mo "hair" + kio "bridge". A reference to the Europeans, who were derogatorily called "red hair devils" (describes the bridge that was built by the Europeans). |
| NS17 CC15 | Bishan | Chinese | The pinyinization of Peck San (碧山), derived from Peck San Theng (碧山亭), a Chinese cemetery (describes the land use in the area). |
| NS18 | Braddell | English | Named after Thomas Braddell (eponymous), an Irish lawyer and the first Attorney General of Singapore. |
| NS19 | Toa Payoh | Hokkien Malay | Toa "big" (Hokkien) + payoh (a loanword from the Malay word for swamp, paya). The place was mainly forest and swamp (describes the type of swamp formerly in the area). |
| NS20 | Novena | Latin | Named after the Novena Church located along Thomson Road (associative: church). |
| NS21 DT11 | Newton | English | Named after Howard Vincent Newton (eponymous), the Assistant Municipal Engineer in late nineteenth-century Singapore. |
| NS22 TE14 | Orchard | English | Named after the numerous nutmeg farms, pepper farms, spice gardens, and fruit orchards that covered the area since the 1830s (describes the trees and plants grown in the area). |
| NS23 | Somerset | English | Named after a county in England (borrowed from a British county). |
| NS24 NE6 CC1 | Dhoby Ghaut | Hindi | Dhoby "laundry" + ghaut "steps along the bank of a river". Named after laundry activities that happened here from the 1830s (describes the land use in the area). |
| EW13 NS25 | City Hall | English | Named to celebrate Singapore being granted city status (occurrent: historic event). |
| EW14 NS26 | Raffles Place | English | Formerly known as Commercial Square, it was renamed after Sir Stamford Raffles (eponymous), who was regarded as the founder of Singapore, in 1858. |
| NS27 CE2 TE20 | Marina Bay | English | Named after the water body it surrounds (associative: water body). |
| NS28 | Marina South Pier | English | Named after the Marina South Pier, which is nearby (associative: water body). |

## Appendix B  Downtown Line

**Table A2.** Analysis of MRT Stations on the Downtown Line.

| Station No. | Station Name | Language | Naming Practice (Strategy Is <u>Underlined</u>) |
|---|---|---|---|
| DT1 *BP6* | Bukit Panjang | Malay | Bukit "hill" + panjang "long" (<u>describes</u> the type of hill in the area). |
| DT2 | Cashew | English | Named after the nearby Cashew Road (<u>associative</u>: road). |
| DT3 | Hillview | English | Named after the nearby Hillview Road (<u>associative</u>: road). |
| DT5 | Beauty World | English | Named after the historic Beauty World, an amusement park and market that once stood in this area (<u>associative</u>: amusement park and market). |
| DT6 | King Albert Park | English | Named after a nearby road of the same name (<u>associative</u>: road). |
| DT7 | Sixth Avenue | English | Named after a nearby road of the same name (<u>associative</u>: road). |
| DT8 | Tan Kah Kee | Hokkien | Named after Tan Kah Kee (<u>eponymous</u>), who founded Hwa Chong Institution, where the MRT station is located. |
| DT9 CC19 | Botanic Gardens | English | Named after the nearby Singapore Botanic Gardens (<u>associative</u>: tourist attraction). |
| DT10 <u>TE11</u> | Stevens | English | Named after the nearby Stevens Road (<u>associative</u>: road). |
| DT11 NS21 | Newton | English | Named after Howard Vincent Newton (<u>eponymous</u>), the Assistant Municipal Engineer in late nineteenth-century Singapore. |
| DT12 NE7 | Little India | English | Named after the ethnic neighborhood of Little India (<u>describes</u> land use in the area), which the station is located within. |
| DT13 | Rochor | Malay | One account of the origins of Rochor is based on Chinese folklore claims that Rochor is derived from Wu Zhu, the founder of the Heaven and Earth Society, who wanted to overthrow the Qing Dynasty. Wu sailed up the Rochor River and settled on its banks. The second links the name to Cho Ah Chi, a carpenter who was on Raffles' expedition to Singapore in 1819. Cho was a member of a triad and set up a branch in Rochor, giving the district its name (<u>legends and anecdotes</u>). |
| DT14 EW12 | Bugis | Malay | Named after the Bugis people (<u>borrowed</u> from the name of a local ethnic group), who, in 1820, fled Riau and settled in Singapore. |
| DT15 CC4 | Promenade | English | Named after the MRT station's geographical proximity to Marina Promenade (<u>associative</u>: promenade). |
| DT16 CE1 | Bayfront | English | Named after Bayfront Avenue, where the station is located (<u>associative</u>: road/avenue). |
| DT17 | Downtown | English | Named after the Downtown Core Planning Area and Downtown Commercial District, which the station serves (<u>associative</u>: vicinity). |
| DT18 | Telok Ayer | Malay | Telok "bay" + ayer "water". Telok Ayer was situated along the old shoreline of Singapore, hence giving its name (<u>associative</u>: water body). |
| DT19 NE4 | Chinatown | English | Named after the ethnic district of Chinatown (<u>describes</u> land use in the area), which the station is located within. |
| DT20 | Fort Canning | English | Named after Fort Canning Hill, located to the north of the station (<u>associative</u>: hill). |
| DT21 | Bencoolen | English | Named after Bencoolen Street (<u>associative</u>: street). |
| DT22 | Jalan Besar | Malay | Jalan "road" + besar "big/wide" (<u>describes</u> the type of road formerly in the area). |
| DT23 | Bendemeer | Unknown | Despite being located underneath the road of Kallang Bahru, the station is named after the nearby Bendemeer Road (<u>associative</u>: road). |

**Table A2.** *Cont.*

| Station No. | Station Name | Language | Naming Practice (Strategy Is <u>Underlined</u>) |
|---|---|---|---|
| DT24 | Geylang Bahru | Malay | Despite being located at Kallang Bahru, the station is named after the junction Geylang Bahru (<u>associative</u>: junction). |
| DT25 | Mattar | Unknown | Named after a nearby road, Mattar Road (<u>associative</u>: road). |
| DT26 CC10 | MacPherson | English | Named after Lieutenant Colonel Ronald MacPherson, the first Colonial Secretary of the Straits Settlements, which Singapore was a part of, in 1867 (<u>eponymous</u>). |
| DT27 | Ubi | Malay | Named after ubi kayu (tapioca), an important staple during the Japanese occupation, commonly grown around the area (<u>describes</u> a crop grown in the area). |
| DT28 | Kaki Bukit | Malay | Named from the Kaki Bukit industrial area, which the station serves (<u>associative</u>: industrial estate). |
| DT29 | Bedok North | MalayEnglish | Named after Bedok North Road, where the station is located (<u>associative</u>: road). |
| DT30 | Bedok Reservoir | MalayEnglish | Named after the nearby Bedok Reservoir (<u>associative</u>: water body). |
| DT31 | Tampines West | MalayEnglish | While there is no indication in the address of the station that it is in Tampines West, the station is located near to structures like the Tampines West Community Clu and Fitness 45 Tampines West club, among others (<u>associative</u>: structures). |
| DT32 EW2 | Tampines | Malay | Named after Tempinis trees, which grew in the area (<u>describes</u> trees grown in the area). |
| DT33 | Tampines East | MalayEnglish | While there is no indication in the address of the station that it is in Tampines East, the station is located near to structures like the Tampines East Community Club and Tampines East Zone 2 Residents Committee (<u>associative</u>: structures). |
| DT34 | Upper Changi | MalayEnglish | Named after Upper Changi Road (<u>associative</u>: road), where the station is located. |
| DT35 | Expo | English | Named after Singapore Expo, which is located beside it (<u>associative</u>: convention hall). |

Key: EW = East West Line. NS = North South Line. NE = North East Line. CC; CE = Circle Line. DT = Downtown Line. TE = Thomson East Coast Line (under construction). JE = Jurong Region Line (under construction). CR = Cross Island Line (under construction). BP = Bukit Panjang Light Rail Transit.

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
