# Peer review of "Then and Now: A Comparative Historical Toponomastics Analysis of Station Names in 2 of Singapore’s Mass Rapid Transit (MRT) Lines"

_urbansci, doi:10.3390/urbansci4030037_

Round 1
Reviewer 1 Report
The manuscipt provides new and interesting knowledge about the naming of public transport. The analysis is sound, and the classification principles are presented thoroughly in relation to the former studies.
In describing the methodology, the methods applied could be presented in more detail. It could be stated what the "(socio)linguistic and historical toponomastics approach" actually stand for, preferably using more international references. Here, e.g. The Oxford Handbook of Names and Naming (ed. Hough, 2016) would be useful. Moreover, the choice of the material needs some further clarification. Why it was especially these two MRT lines chosen? And could the results look a bit different if some other lines would have been studied?
To conclude, the title of the paper should be rethinked. Since it is only station names examined, this would be beneficial to include to the title.
Author Response
In Section 5, Methodology, I have, as you have suggested, incorporated approaches from the Oxford Handbook of Names and Naming into the methodology. I have also explained at the beginning of the paragraph why the North South and Downtown Lines are chosen (as to whether the results would look different if other lines are selected, this is a topic that requires further study).
Additionally, thank you for your pointer on the title. I have changed the title to better reflect this.
Reviewer 2 Report
Thank you for the opportunity to review this paper.
Living in a time when urban place naming is quite neglected by many urbanists and other social scientists I think this paper will be well received by urban toponymists.
I have attentively read this paper and I have to make only a major observation. Even the paper is very well grounded in a good method and brings excellent results, I think that authors could position a bit better the paper in the international literature. I would suggest the authors to introduce a literature review section named 'Urban Place Naming and Historical Change', in which issues of place naming to be enlarged, mentioning other important toponymy works of Azaryahu (1996), Alderman (2002), Reuben-Redwood (2011), Cretan and Matthews (2016) etc positioning better this study on international toponymy at large. Thus going from more general issues (but not limited only to street naming but also to school naming - see a study in Journal of Historical Geography in 2019 or to other institutional naming) to particular aspects, authors could better position their study at international level. For example, it could be mentioned that place naming included commodification at different levels not only the Dubai example given in this paper, there are even extensions to sports names as footbal club's names, see for instance an article about how a club's name in the city of Timisoara, Romania brought tensions among fandom in the city. That study was published in 2019 in one of Urban Geography journal's special issue on urban naming practices and could be mentioned as an angle of research in current urban toponymy.
Also, neoliberalism might be highlighted by the authors of this paper as a current pressure on urban toponymy worldwide.
Literature review section could include some of the sentences presented now by the authors in the introduction and should not be longer than four paragraphs.
The rest of the paper looks perfect to me.
Author Response
Dear Reviewer,
Thank you for your kind suggestions. I have sought to incorporate most of your pointers and sources in the Introduction rather than having a separate section for this altogether. To this end, I have included Azaryahu (1996), Alderman (2002), and Cretan and Matthews (2016). Rose-Redwood's 2011 piece is already inside my references for the article. I have also worked in the piece on the contestations of football club naming (it's a pretty interesting topic!)
Thank you for your time once again and hope this works :)
Reviewer 3 Report
It was delightful to read this well structured and tailored piece which expands on the relationship between public transport and naming policies, reflecting Singapore's culture, history, and geopolitical realities.
My comments relate mainly to the Introductory section, which is too tight and bypasses a few important lens which can contribute to a better understanding of the importance of the current piece:
The literature about toponyms and transportation in neo-liberal commercialised contexts is very scant, and I think that the author should be more aware of this situation in terms of historiography: a reference to two of three localities in this context, such as the naming policy of Dubai's metro stations -- https://www.rta.ae/links/NamingRights/criteria.html/
https://wam.ae/en/details/1395302843442
-- can hence to the contribution of the present piece.
A note on toponymic policies in multilingual societies could be also usable in this state of study context, for some key references see:
* Bigon L., and A.J. Njoh, 2015. "The Toponymic Inscription Problematic in Urban Sub-Saharan Africa: From Colonial to Postcolonial Times", Journal of Asian and African Studies, 50, 1: 25-40
* Jones, Rhys and Peter Merriman, 2009. "Hot, Banal and Everyday Nationalism: Bilingual Road Signs in Wales," Political Geography, 28: 164-173.
* Raento P and Watson C, 2000, "Gernika, Guernica, Guernica? contested meanings of a Basque place" Political Geography, 19: 707-736
Additional comments:
Only on p. 8 the riddle why the authors have not treated the toponyms of Singapore’s sixth and newest MRT line are solved: maybe to clarify this earlier, at the introduction.
It could be great if the authors will provide a visual records of one or two signposts of the MRT in situ with explanatory captions and a note on the actual signage policy as well (the materiality of the signs beyond the naming per se).
I must admit that the last sentence of the article is most interesting and raises many questions: "This would provide insights on commuters opinions towards naming policies of MRT stations (given that Singaporeans are now actively involved in naming newer lines) and their linguistic sensitivity about the choice of MRT names." While investigating the perception Singapore citizens have of their MRT system's toponyms is (unfortunately!) beyond the scope of this research; maybe the authors could at least tell us in a few words how or in which ways the Singaporeans are actively involved in the naming process.
Author Response
Dear Reviewer,
Thank you for your time and helpful suggestions. I have incorporated most of the literature you have kindly suggested into the introduction to lend a more "international" and "holistic" approach to the introduction/literature review.
As for your point on why the North South and Downtown Lines are chosen, I have included the rationale in the Introduction and also repeated it in the Methodology.
Lastly, the conclusion has also been edited to explain how Singaporeans are more involved in the naming process.
Hope this helps and works!